# Studies Using Mutant Strains of *Azospirillum brasilense* Reveal That Atmospheric Nitrogen Fixation and Auxin Production Are Light Dependent Processes

**DOI:** 10.3390/microorganisms11071727

**Published:** 2023-06-30

**Authors:** Alexandra Bauer Housh, Randi Noel, Avery Powell, Spenser Waller, Stacy L. Wilder, Stephanie Sopko, Mary Benoit, Garren Powell, Michael J. Schueller, Richard A. Ferrieri

**Affiliations:** 1Missouri Research Reactor Center, University of Missouri, Columbia, MO 65211, USA; alibauer1109@gmail.com (A.B.H.); rlnhmh@umsystem.edu (R.N.); avery.snyder00@gmail.com (A.P.); spenserwaller@gmail.com (S.W.); wildersl@missouri.edu (S.L.W.); sa207945@atsu.edu (S.S.); mvbf4w@mail.missouri.edu (M.B.); gmp722@health.missouri.edu (G.P.); schuellerm@missouri.edu (M.J.S.); 2Chemistry Department, University of Missouri, Columbia, MO 65211, USA; 3Interdisciplinary Plant Group, University of Missouri, Columbia, MO 65211, USA; 4Division of Plant Science & Technology, University of Missouri, Columbia, MO 65211, USA; 5School of Natural Resources, University of Missouri, Columbia, MO 65211, USA; 6Department of Biochemistry, University of Missouri, Columbia, MO 65211, USA

**Keywords:** *Azospirillum brasilense*, biological nitrogen fixation, microbial auxin biosynthesis, ATP biosynthesis, micronutrient uptake, light stimulation

## Abstract

As the use of microbial inoculants in agriculture rises, it becomes important to understand how the environment may influence microbial ability to promote plant growth. This work examines whether there are light dependencies in the biological functions of *Azospirillum brasilense*, a commercialized prolific grass-root colonizer. Though classically defined as non-phototrophic, *A. brasilense* possesses photoreceptors that could perceive light conducted through its host’s roots. Here, we examined the light dependency of atmospheric biological nitrogen fixation (BNF) and auxin biosynthesis along with supporting processes including ATP biosynthesis, and iron and manganese uptake. Functional mutants of *A. brasilense* were studied in light and dark environments: HM053 (high BNF and auxin production), *ipdC* (capable of BNF, deficient in auxin production), and FP10 (capable of auxin production, deficient in BNF). HM053 exhibited the highest rate of nitrogenase activity with the greatest light dependency comparing iterations in light and dark environments. The *ipdC* mutant showed similar behavior with relatively lower nitrogenase activity observed, while FP10 did not show a light dependency. Auxin biosynthesis showed strong light dependencies in HM053 and FP10 strains, but not for *ipdC*. Ferrous iron is involved in BNF, and a light dependency was observed for microbial ^59^Fe^2+^ uptake in HM053 and *ipdC*, but not FP10. Surprisingly, a light dependency for ^52^Mn^2+^ uptake was only observed in *ipdC*. Finally, ATP biosynthesis was sensitive to light across all three mutants favoring blue light over red light compared to darkness with observed ATP levels in descending order for HM053 > *ipdC* > FP10.

## 1. Introduction

Light has been a vital driver for life on Earth, but it has also proved to be threatening to many organisms due to the photodynamic effect [1]. Over time, organisms have developed adaptations to cope with, or even to exploit, light. It is common knowledge that, as autotrophs, light has vital importance to the growth and development of higher plants. What is less known is how light might impact the belowground processes involving the roots and/or the microorganisms that colonize them. In a recent study, it was demonstrated that photoreceptors located within the roots of *Arabidopsis thaliana* could directly perceive certain wavelengths of light (predominantly red light) that was conducted, or “piped”, from shoot-to-root tissues where it activated root-expressed *phyB*, and in turn, up-regulated the HY5 transcription factor [2]. HY5 has a role in root growth, moderating their gravitropism responses [2] and ability to assimilate nitrogen from the soil interface [3]. Light-piping has also been described in bean plants and maize [4,5,6]. If plant roots contain photoreceptors and can receive light stimuli through conduction from the aerial portions of the plant, it becomes of interest whether root-associating plant growth-promoting bacteria (PGPB) and the biological functions associated with their symbiosis can respond to light.

Bacteria of the genus *Azospirillum* are Gram-negative, nitrogen fixing bacteria found in the rhizosphere, which are known for their plant growth promoting effects on grasses and cereals worldwide [7,8,9,10,11,12,13]. When soil and environmental conditions are ideal, *Azospirillum* can promote plant growth, crop yield, and improve overall nitrogen content within the plant [14,15,16,17]. These PGPB are thought to positively impact plant performance due to mechanisms such as their BNF [16] and their ability to produce an important plant developmental hormone, auxin [9,16,18,19,20]. The positive impacts this bacterium has on its host plants has led to the commercialization and use of the bacteria inoculants in certain areas of the world [17,18,20,21].

Like most plants, many microorganisms can be sensitive to light, most notably, photosynthetic species such as cyanobacteria, with the most well-studied photoreceptor in photosynthetic bacteria being a phytochrome called bacteriophytochrome [22]. For example, *Rhodospirillum rubrum*, a phototrophic soil microorganism, has been shown to possess photoreceptors that can render its nitrogenase enzyme light sensitive [23]. Photoreceptors, however, are not restricted to phototrophic bacteria. Phytochromes, a family of red/far-red responsive photoreceptors, are present in both phototrophic and non-phototrophic microorganisms [1]. *Azospirillum brasilense* contains genes for bacteriophytochrome that control carotenoid-independent responses to photodynamic stress [1]. This indicates that although these bacteria are not phototrophic, they are equipped to sense light and respond to it. Currently, there is little understanding of light-sensing and the impact of light stimuli on microbial biological functions within *A. brasilense*. The early work reporting on photodynamic stress within the Sp7 wild-type strain of *A. brasilense* did not show evidence that light impacted its growth performance [1]. However, a more recent study showed evidence that light stimuli inhibited microbe motility [24]. To our knowledge, this is the first time a light dependency study has been conducted to examine the biological functions of *A. brasilense*.

## 2. Materials and Methods

### 2.1. Bacteria Growth

Three functional mutant strains of *Azospirillum brasilense* (HM053, *ipdC*, and FP10) were used in the study and obtained through a material transfer agreement between the Federal University of Paraña (UFPR, Curitiba, PR 174CEP 81531-980, Brazil) and the corresponding author’s institution. The HM053 mutant is a natural mutant of the wild-type strain of *A. brasilense* FP2 (Sp7 ATCC 29145 Nif^+^Sm^r^Nal^r^) screened through its resistance to ethylenediamine (EDA^r^) [25,26]. For the *ipdC* mutant, the indole-3-pyruvate decarboxylase gene (*ipdC*), coding for a key enzyme of the indole-3-pyruvic acid pathway of auxin (indole-3-acetic acid) biosynthesis in *Azospirillum brasilense*, was functionally disrupted in a site-specific manner using a SacB-cassette insertion into the *ipdC* gene of wild-type FP2 (Sp7 ATCC 29145) followed by homologous recombination. The method allowed for the construction of the *ipdC* mutation strain without unwanted sequence changes and relied on the λ Red recombineering method (Direct and Inverted Repeat stimulated excision; DIRex) which works well for generating single point mutations, small insertions or replacements, as well as deletions of any size, in a bacterial gene [27]. The resultant knock-out strain exhibited a significant reduction in auxin biosynthesis to a level of 10% that of the wild-type strain [19]. The FP10 mutant was obtained by N-nitrosoguanidine mutagenesis of the FP2 wild-type strain of *A. brasilense* and isolated by growth on glutamate medium [28].

The functional mutants were grown in liquid NFbHP-lactate medium at pH 6.5 following published procedures [16]. The medium contained 20 mM ammonium chloride (NH_4_Cl) as a nitrogen source and streptomycin antibiotic (80 μg mL^−1^). The cultures were grown in a shaking incubator at 30 °C and 130 rpm in 10 mL volume, then re-propagated into 40 mL volume approximately 24 h prior to starting the tracer administration. The bacteria were grown under two treatment types: 24 h light and 24 h dark. During the light experiments, bacteria were propagated directly under a 30 cm × 30 cm red-blue LED light panel intended for indoor plant growth (200 μmol m^−2^ s^−1^ total intensity) that was positioned over the shaking incubator. Lights were left on for the entirety of the bacteria growth period. Dark experiments involved propagation of the bacteria in the dark, handled under indirect light, and grown in the shaking incubator at 30 °C while shielded from any ambient light.

### 2.2. Plant Growth

Maize kernels (Hybrid 8100) from Elk Mound Seed Co. (Elk Mound, WI, USA) were dark germinated at room temperature for two days on sterilized paper towels wetted with sterile water in a petri dish. Seedling’s roots were inoculated with isolated bacteria strains suspended in sterile water for 3 h prior to transplanting to a growth pouch wetted with sterile Hoagland’s basal salt solution for approximately one week. They were then transferred to eight-inch plastic cones filled with Turface™ expanded clay matrix (Profile Products, LLC., Buffalo Grove, IL, USA) where the bottom portion of the cone was immersed in de-ionized water. Nutrients were introduced as Hoagland’s solution every three days. Growth conditions consisted of 12 h photoperiods, 500 µmol m^−2^ s^−1^ light intensity, and temperatures of 25 °C/20 °C (light/dark) with humidity at 70–80% for two weeks. For BNF studies, roots were removed from the Turface™ and subjected to the acetylene reduction assay described below for measuring microbial nitrogenase activity. A subset of each root mass collected was also subjected to drop plate analysis to measure the extent of microbial root inoculation.

### 2.3. Acetylene Reduction Assay (ARA)

ARAs were conducted to examine functionality of the nitrogenase enzyme under light and dark growth conditions. In this approach, acetylene gas is used to measure nitrogenase activity by its ability to reduce acetylene to ethylene. For details on the ARA, see the Appendix A.

The bacteria were grown in association with maize roots, which were harvested, weighed (typically ~1 gfw) and placed in a 500-mL Mason jar where the top was modified with a gas sampling port to introduce the acetylene gas and withdraw samples later for gas chromatography analysis (Appendix A). For details on gas chromatography analysis, see the Appendix A. Ethylene data from this analysis were normalized to CFUs per gram of root tissue as determined by Drop Plate Assays in each experiment. For details on the Drop Plate Assay, see the Appendix A.

For light sensitivity studies, the Mason jars were exposed to full-spectrum white light using a white LED light panel (200 µmol m^−2^ s^−1^ intensity) or individual red LED or blue LED light panels (800 µmol m^−2^ s^−1^ intensities). For full darkness measurements, Mason jars were wrapped in aluminum foil. For consistency, all experimental measurements were carried out at 30 °C.

### 2.4. Microbial ^59^Fe Uptake and Metabolic Transformation

Radioactive ^59^Fe^+3^ and ^59^Fe^2+^ (t_1/2_ = 44.5 days) were purchased in 1 mCi doses (equivalent to 37 MBq) from Perkin Elmer Life Sciences (Akron, OH, USA). Liquid cultures of bacteria were grown for 24 h in liquid NFbHP-lactate media prior to the start of a ^59^Fe radiotracer study. On the day of an experiment, the cultures were centrifuged, washed with HyPure™ sterile deionized water (Cytiva HyClone Laboratory, Logan, UT, USA), and combined to yield a concentrated pellet of bacteria that was resuspended in 11 mL of HyPure™ water which had a slightly acidic pH of 5.5. A UV reading was taken at 600 nm (O.D._600_) using a 2 mL aliquot of the 11 mL sample. Of the 9 mL bacteria sample remaining, 9 aliquots of 1 mL were placed into 15 mL labeled Falcon tubes. Dosing with either ^59^Fe^3+^ or ^59^Fe^2 +^radiotracers as chloride salts was accomplished by adding radiometal diluted in HyPure™ sterile water to achieve 20 µCi doses (equivalent to 7.4 × 10^5^ Bq) per 100 µL volume of water. Each dose possessed less than 1 nmole of ^nat^.Fe carrier, bringing the concentration to 100 nM. Start times were recorded for radioactive decay correction back to common zero times. An aliquot of each diluted solution of ^59^Fe radiotracer was counted in the NaI(Tl) gamma well counter and corrected for volume and detector efficiency back to a value for activity administered to each Falcon tube. Samples were incubated on a Unico TTR-2000 test tube rocker (United Products & Instruments, Inc., Dayton, NJ, USA) for 1, 3, and 5 h time points either under combined (red/blue) LED lights or shrouded in an aluminum foil blanket for dark studies. All radiotracer uptake data were normalized to a standard O.D._600_ value of 2.00.

At each time point, the appropriate samples were centrifuged, washed with sterile HyPure™ water three times, and re-suspended in 1 mL of 1 M HCl (pH << 1). An aliquot (100 µL) of each was removed for activity measurements in the NaI(Tl) gamma well counter for a measure of activity before extraction. The remainder of the washed bacteria were cell-disrupted for 2 min under 100% amplitude to burst open the cells for ^59^Fe extraction. Once complete, samples were centrifuged for 2 min and the supernatant removed; 100 µL of the supernatant was removed for immediate counting in the NaI(Tl) gamma well detector, while the remaining 900 µL was used for iron oxidation state speciation via ion chromatography (Appendix A). For details on the ion chromatography analysis, see the Appendix A.

During ion chromatography, the ^59^Fe^3+^ and ^59^Fe^2+^ peaks were collected after separation and the amount of radioactivity in each fraction was measured with the NaI (Tl) gamma well detector. The percent tracer uptake was calculated for each bacterial functional mutant under light and dark conditions at 1, 3 and 5-h intervals. Additionally, the distribution of ^59^Fe^3+^ and ^59^Fe^2+^ ions present was measured to determine if transformation of the oxidation state of the tracer administered remained the same or was altered by our analytical workup of samples, or by the presence of the bacteria functional mutants. We note that while ferrous iron is soluble and relatively stable under anaerobic acidic conditions, under pH neutral aerobic conditions, ferrous iron can oxidize rapidly to its ferric form and be hydrolyzed to insoluble ferric hydroxide Fe(OH)_3_. Because our radiotracer uptake studies were conducted under mildly acidic conditions followed by strongly acidic conditions during analytical workup, we did not expect environmental oxidation of ferrous iron to be a problem. Even so, ^59^Fe^2+^ stability was examined without the presence of bacteria under our experimental protocol to verify this fact. Furthermore, prior studies on the hydrolysis of ferric chloride in dilute solutions showed that the rate at which ferric hydroxide is precipitated from such solutions decreases as the iron concentration decreases, reaching zero at a concentration of 2 nM [29]. This process is also pH-dependent and can be important at pH values above 6.5. As our studies were conducted in HyPure™ sterile water at pH 5.5, and at a concentration of 1 nM iron, it is unlikely that radiotracer hydrolysis and/or precipitation as ferric hydroxide was a factor to consider.

### 2.5. Luciferase Chemiluminescence ATP Assay

ATP levels were determined using BacTiter-Glo™ Microbial Cell Viability Assay Reagent (Promega, Madison, WI, USA). This was a luciferase-based assay and the ATP level was determined by measuring luminescence levels compared to ATP standards [30]. Measurement of ATP concentrations within the bacteria cells was performed using a published method of resuspending the bacteria pellet in 500 µL of deionized H_2_O and boiling the solution for 10 min to break down the cells [31]. For details on the experimental methods, see the Appendix A. All data were normalized to a standard O.D._600_ value of 2.00.

### 2.6. Spectrophotometric Auxin Assay

Colorimetric analysis of auxin was performed using Salkowski reagent [32,33]. When carrying out the assay, 200 µL of bacterial culture was mixed with 300 μL of Salkowski reagent and kept in the dark for at least 30 min at room temperature. Additional samples from the same culture were centrifuged where the pellet was separated washed once, resuspended in 500 µL of H_2_O, and boiled for 10 min to release cellular auxin. Two hundred µL aliquot was mixed with 300 µL of Salkowski reagent and kept in the dark for at least 30 min at room temperature. Sample absorbances were measured at 536 nm using a spectrophotometer (Evolution 201 UV/VIS, ThermoFisher Scientific Inc., Waltham, MA, USA). For quantitative analysis, serial dilutions from 0 to 50 mg/L of auxin (Sigma Aldrich, St. Louis, MO, USA) were prepared and used as standards and the bacteria content in all analyses were normalized to a standard O.D._600_ value of 2.00.

### 2.7. Bacteria ^52^Mn Uptake

Radioactive ^52^Mn^2+^ (t_½_ = 5.59 d) decays 29.6% by positron emission. Positron annihilation resulting in two coincident gamma rays at 511 keV energy makes this radioisotope ideal for tracking plant uptake of manganese via gamma counting. For these studies, we purchased a 1 mCi dose of ^52^Mn^2+^ (equivalent to 37 MBq) from the University of Wisconsin [34]. Details on how this isotope was produced are found in the Appendix A. Upon receipt of the radioisotope, the dose was pH neutralized since it was shipped in dilute HCl and it was diluted further with HyPure™ sterile deionized water for administration to liquid cultures of bacteria isolates following the same procedures described for the ^59^Fe studies. Samples were incubated on a rocking platform for 1, 3, and 5 h time points either under LED lights or wrapped in aluminum foil for dark studies.

### 2.8. Statistical Analysis

Data were subjected to one-way analysis of variance (ANOVA) in R using SigmaPlot 14.5. Tukey’s HSD test was used for post hoc correction of comparisons across the treatment conditions (i.e., light vs. dark and red vs. blue light) at a significance level of *p* < 0.05. The ^59^Fe^3+/2+^ and ^52^Mn^2+^ allocation data were also analyzed by Principal Component Analysis (PCA) using XLSTAT software version 2020.3 (Addinsoft Inc., New York, NY 10001, USA).

## 3. Results and Discussion

### 3.1. Light Dependencies of BNF and Its Supporting Processes in A. brasilense

To investigate whether the activity level of the bacterial nitrogenase enzyme, responsible for BNF capacity, exhibits a light dependency, an ARA was performed with mutant bacteria inoculated maize roots in both full spectrum white light and darkness conditions (Figure 1A–C), as well as in red light and blue light conditions (Figure 1D,E). Regardless of light vs. dark conditions, the ethylene levels slowly increased over the longer root incubation time. This is expected based on natural root emissions of their own ethylene while incubating in the closed jar, as is also measured on non-inoculated maize roots [19]. Natural root ethylene emissions were more obvious with FP10, a non-BNF strain where the slow rise in ethylene over time was due to this process. ARA data were normalized for the total root mass in each sample and average microbial content determined from drop plate assays. However, we did not correct for the ethylene contributions derived from natural root emissions because they were miniscule [19] relative to the levels of ethylene generated by the ARA and did not confound interpretation of microbial light dependencies.

It is clear from the light/dark ARA studies that the BNF-capable bacteria mutants, HM053 and *ipdC*, showed a significant light dependency in terms of ethylene production indicating that the nitrogenase enzyme in both these strains was more active in the presence of light than in darkness. HM053, which fixes much more nitrogen than *ipdC*, had values much greater than those observed in the latter especially while exposed to light, producing upwards of 1000 µmol ethylene per gram fresh weight of root (µmol ethylene gfw^−1^) after 6 h compared to the 90 µmol ethylene gfw^−1^, respectively. Strikingly, the FP10 mutant, which does not contain an active nitrogenase gene, did not exhibit a light dependency. This seems to point to light sensitivity of the nitrogenase enzyme, or at least the BNF capacity of the bacteria, in *A. brasilense*.

Following the white light studies, comparisons were made between red (660 nm) and blue light (445 nm) stimulation of nitrogenase activity in BNF-capable mutants of HM053 and *ipdC*. Here, it was noted that HM053 and *ipdC* both favored red light for stimulating their nitrogenase enzymes. Additionally, while the wavelength specific light studies were performed at higher light intensities (i.e., 4 times that of the full spectrum white light source) the nitrogenase enzyme in HM053 showed a 6-fold higher level of activity in red light than in the full-spectrum white light after correcting for this intensity difference. On the other hand, blue light stimulation showed approximately the same level of enzyme activity as the full spectrum white light after intensity adjustments were made. Applying the same adjustments for differences in light intensity to the *ipdC* bacteria, we found that red light stimulation boosted nitrogenase activity 17-fold relative to full-spectrum white light, while blue light stimulation resulted in approximately the same level of enzyme activity as the full-spectrum white light.

Light sensitivity in the nitrogenase function has been observed in oceanic photosynthetic bacteria, *Crocosphaera watsonii*, which, upon exposure of the bacteria to blue light, caused a decrease in nitrogenase activity [35]. In another photosynthetic bacteria species, *Rhodopseudomonas palustris*, a nitrogenase variant exists in which two amino acid substitutions were observed in the FeMo protein allowing the enzyme to reduce CO_2_ to CH_4_ [36]. In these bacteria, the nitrogenase-variant enzyme was seen to produce methane only in the presence of light and more methane as the light intensity increased with the limitation that the highest intensity explored was 60 µmol m^−2^ s^−1^ [35]. While light influences on nitrogenase function in non-photosynthetic bacteria have not been explored; light clearly is shown in these instances to influence the BNF capacity of these organisms. *A. brasilense* functional mutants did not exhibit BNF inhibition with exposure to light, as noted for *Crocosphaera watsonii*, suggesting that light influences on BNF activity are perhaps not conserved between photosynthetic and non-photosynthetic bacteria species.

Our interest in ATP biosynthesis as a supporting process is due to BNF being a multi-electron redox process carried out by the nitrogenase enzyme in all diazotrophic species which serves to reduce atmospheric nitrogen to a biologically usable form, NH_3_ [37], and is driven by ATP as its chemical energy source. BNF requires at least 16 Mg-ATP molecules to reduce a single dinitrogen molecule [38]. Because nitrogen fixation is so energetically costly, its regulation is essential for balancing BNF with growth performance of any diazotrophic microorganism. Nitrogenase activity is regulated transcriptionally through NifA, the transcriptional activator [9], and post-translationally involves dinitrogenase reductase-activating glycohydrolase together with the P_II_ protein GlnZ [39,40,41].

Many phototrophic microorganisms such as algae and cyanobacteria carry out photophosphorylation, a light dependent process involving the phosphorylation of ADP making ATP. However, to the best of our knowledge, nothing is known regarding non-phototrophic microorganisms such as *A. brasilense* and whether its ability to biosynthesize ATP is light-dependent.

Comparative measurements using the luciferase chemiluminescence assay were conducted in red and blue light conditions using the HM053, *ipdC*, and FP10 mutant strains (Figure 2). All three strains exhibited strong light dependencies in their cellular ATP concentration for both red and blue light when compared to darkness, although in all cases, there was a strong preference for blue light. Although *A. brasilense* is non-phototrophic, this observation is consistent with past work demonstrating that blue light was most efficient at promoting photophosphorylation in phototrophic microorganisms [38].

Nitrogenase is a two-protein enzyme where one protein contains an iron center and the other an iron–molybdenum center [42]. Iron protein is the only known electron donor to support BNF, and thus, is vital for its function. The predominant oxidation state of the iron in the nitrogenase iron protein is Fe^2+^ [42].

In the present work, we explored whether microbial iron uptake, as Fe^3+^ and Fe^2+,^ exhibited certain light dependencies. Using radioactive ^59^Fe in its ferric ^59^Fe^3+^ oxidation state, or in its ferrous ^59^Fe^2+^ oxidation state, we were able to measure microbial uptake of tracer over a 5 h incubation period as a function of light versus darkness for HM053, *ipdC*, and FP10 mutant strains. As noted earlier, the light studies used a large red/blue light panel that illuminated the entire culture tube rocker assembly with red/blue light of equal intensities. Results in Figure 3A–C show the uptake curves for ^59^Fe^3+^ in the three bacteria strains. Interestingly, HM053 showed a strong light dependency for ^59^Fe^3+^ uptake where more ferric iron was taken up in darkness than in light. Neither *ipdC*, nor FP10 showed a light dependency for ^59^Fe^3+^ uptake. Contrary to this, HM053 and *ipdC* showed a light dependency for ^59^Fe^2+^ uptake where ferrous iron uptake was significantly higher in light than in darkness (Figure 3D,E). This observation correlates well with ARA data in Figure 1, and the fact that Fe^2+^ is vital to the function of the nitrogenase enzyme in the HM053 and *ipdC* strains. However, our HM053 ARA data showed much higher levels of nitrogenase activity than *ipdC*, but the data in Figure 3 indicate that *ipdC* acquired more ^59^Fe^2+^ over the 5 h incubation period than HM053. This suggests that Fe^2+^ likely plays other roles in the growth and function of these microorganisms. Consistent, too, with ARA data FP10, the BNF deficient strain did not show a light dependency for ^59^Fe^2+^ uptake (Figure 3F).

Upon surveying the early literature that addressed metal nutrient uptake and metabolic transformation within live cells of rhizospheric microorganisms, and particularly *Azospirilla*, we found the number of published works scarce [43,44,45,46]. It was not until later applications of Mössbauer spectroscopy (reviewed in [47]) that a greater understanding of microbial demands for iron came to light [48,49,50,51,52]. In this collection, studies using wild-type Sp245 *A. brasilense* grown aeroponically in cultures containing ^57^Fe^3+^-nitrilotriacetate complex showed that live cells reduced 33% of the assimilated Fe^3+^-to-Fe^2+^ over an 18 h period of growth [48]. Similarly, wild-type Sp7 *A. brasilense* bacteria reared under the same conditions reduced 22% of the assimilated Fe^3+^-to-Fe^2+^ [50]. These Mössbauer studies suggested that cellular Fe^3+^ was stored in ferritin-like components which was verified in other work [49,51]. Furthermore, these studies showed that Fe^2+^ existed in highly coordinated forms [48,50].

Given the state of understanding of microbial iron metabolism from this extensive body of Mössbauer literature, we wanted to examine whether light affected the metabolic transformation of ferric iron once the metal was taken up by the bacteria cell (Figure 3G). Consistent with the body of Mössbauer data cited above, our radiotracer results showed that under illumination, all three mutant strains of *A. brasilense* converted a small but consistent 30% of the ^59^Fe^3+^ taken up by the bacteria cells to ^59^Fe^2+^ over the course of the 5 h incubation period. In contrast, when all three mutant strains were kept in darkness, we initially observed a lower level of metabolic transformation of ^59^Fe^3+^-to-^59^Fe^2+^ which amounted to only 15% transformation. However, this level rose sharply to 60% ^59^Fe^3+^-to-^59^Fe^2+^ transformation over the 5 h incubation.

A very different story unfolds when ^59^Fe^2+^ was administered (Figure 3H). Here, we observed a small but relatively steady 5% transformation of ^59^Fe^2+^-to-^59^Fe^3+^ when HM053 or the *ipdC* strains were illuminated, suggesting that their active nitrogenase involved in BNF may rapidly appropriate the assimilated ^59^Fe^2+^ in the iron protein of the enzyme preventing its conversion. When placed in darkness, we found nitrogenase was substantially downregulated as determined by ARA. Under these conditions, we expect less ^59^Fe^2+^ to bind to the iron protein and thus more of it remains free to convert to ^59^Fe^3+^. In fact, at the shortest 30 min timepoint, we observed 70% transformation of ^59^Fe^2+^-to-^59^Fe^3+^ for HM053 and *ipdC* strains in darkness. Supporting this theory, studies using FP10, the BNF deficient mutant, showed a consistent level of 75% transformation of ^59^Fe^2+^-to-^59^Fe^3+^ over time under illumination. Like the systematic trend observed in darkness when ^59^Fe^3+^ tracer was administered, there was a linear change in ^59^Fe^2+^-to-^59^Fe^3+^ transformation with time when ^59^Fe^2+^ was administered to bacteria which decreased from 70% to 50% transformation over the 5 h incubation period. We note that regardless of the initial oxidation state of the ^59^Fe radiotracer administered, when in darkness, all bacteria strains appeared to have a larger portion of cellular ^59^Fe^2+^ over time.

### 3.2. Light Dependencies of Auxin Biosynthesis and Its Supporting Processes in A. brasilense

One of the best attributes of *A. brasilense* as an agricultural inoculant lies in its ability to biosynthesize the plant relevant hormone auxin [16,19]. Auxin is best known for its diverse roles in regulating developmental and cellular processes of higher plants impacting axis formation and patterning during post-embryogenesis, vascular elongation, leaf expansion, inflorescence, tropism, and apical dominance [53]. It is also especially important in regulating root development [54,55], where it can cause extensive lateral root patterning and root hair formation. Because of its diverse nature in regulating host growth and development, light dependency on auxin biosynthesis by the *A. brasilense* microorganism was of interest. Using a spectrophotometric assay, we were able to quantify both the total auxin content contained within the bacteria cells and their liquid growth culture (Figure 4A), as well as isolate the cellular and liquid culture components to calculate cellular auxin excretion as a relative percentage of the total content (Figure 4B).

Like the ^59^Fe radiotracer uptake studies, we utilized the same large high intensity red/blue LED light panel to illuminate the liquid culture vials inside the shaking incubator. Our results show that HM053 and FP10 bacteria strains exhibit a significant light dependency when comparing light versus dark condition values. In previous studies, both strains have been reported to biosynthesize auxin at rates of 13.4 ± 0.9 molecules s^−1^ cell^−1^ and 7.0 ± 0.4 molecules s^−1^ cell^−1^, respectively, as determined by our direct radiotracer assay [19], while the *ipdC* auxin deficient mutant strain was estimated to be approximately 10% that of the lower rate. Consistent with earlier findings, the *ipdC* strain did not show a light dependency when comparing the total auxin content between light and dark conditions. Likewise, the relative percent cellular excretion of auxin was the same across all bacterial strains of *A. brasilense* examined and did not appear to change when measurements were conducted in light versus darkness. Thus, we can conclude that the process of microbial cellular auxin excretion is diffusional and likely driven by differences between the physical/chemical properties of the inner cellular and extracellular matrices.

Tryptophan is a key aromatic amino acid precursor in auxin biosynthesis and derives from the shikimate pathway. The shikimate pathway consists of seven sequential enzymatic steps and begins with an aldol-type condensation of two phosphorylated active compounds, the phosphoenolpyruvic acid (PEP), from the glycolytic pathway, and the carbohydrate D-erythrose-4-phosphate, from the pentose phosphate cycle, to give 3-deoxy-D-arabino-heptulosonic acid 7-phosphate (DAHP) (Figure 5).

The seven enzymes that catalyze the pathway are known: 3-deoxy-D-arabino-heptulosonate-7-phosphate synthase (DAHPS; EC 4.1.2.15, now EC 2.5.1.54), 3-dehydroquinate synthase (DHQS; EC 4.2.3.4), 3-dehydroquinate dehydratase/shikimate dehydrogenase (DHQ/SDH; EC 4.2.1.10/EC 1.1.1.25), shikimate kinase (SK; EC 2.7.1.71), 5-enolpyruvylshikimate 3-phosphate synthase (EPSPS; EC 2.5.1.19), and chorismate synthase (CS; EC 4.2.3.5). Three of the seven enzymatic steps within the shikimate pathway rely on Mn^2+^ as a cofactor [56,57]. Hence, a high microbial auxin-producing capacity could lend itself to higher turnover of the shikimate pathway with commensurate higher demands for Mn^2+^ and even higher demands for ATP. The latter comes into play in the fifth step of the shikimate pathway. Here, the shikimate kinase enzyme catalyzes the phosphorylation of the shikimic acid by ATP producing shikimic acid 3-phosphate and ADP. We have already shown that microbial ATP biosynthesis in *A. brasilense* was light-dependent and was up-regulated across the red and blue light spectrum but favored blue light conditions.

Here, we examined whether the light dependency observed for auxin biosynthesis would manifest in a similar light dependency in microbial Mn^2+^ uptake. Using radioactive ^52^Mn^2+^ we were able to measure bacterial uptake of this micronutrient over time as a function of exposure to light versus darkness using the same functional mutants of *A. brasilense*: HM053, *ipdC*, and FP10 (Figure 6A–C).

Surprisingly, the two high auxin-producing mutants HM053 and FP10 did not show the expected light dependency in their ability to assimilate ^52^Mn^2+^. Consistent with a diffusional micronutrient exchange between the growth matrix and the microbial cell, both mutants showed increased uptake of radiotracer over time. However, FP10, which has a lower rate of auxin biosynthesis than HM053 [19] consistently assimilated more ^52^Mn^2+^ than HM053 over the different incubations timepoints examined. Even more surprisingly, the *ipdC* auxin deficient mutant did exhibit a strong light dependency in its ability to assimilate ^52^Mn^2+^ and assimilated much more ^52^Mn^2+^ in darkness than in light. Altogether, these results strongly suggest that microbial Mn^2+^ demands are not directly aligned with their auxin-producing capacity as originally hypothesized. Of course, the role of Mn^2+^ in microbial function likely translates beyond what we described for the shikimate pathway and auxin biosynthesis. More studies are needed to better understand its roles in microbial biology.

Considering that there were interesting light dependencies for ^59^Fe^3+^, ^59^Fe^2+^, and ^52^Mn^2+^ across the different functional mutants of *A. brasilense*, it was of interest to subject the data to Principal Component Analysis across treatment types to elucidate trends (Figure 7). Here, the information included in our uptake measurements were represented by feature vectors (F1 and F2) representing 56.76% and 43.24% of the information embedded in the ^59^Fe^3+^ and ^59^Fe^2+^ data comparison (Figure 6A), and 78.29% and 21.71% of the information embedded in the ^59^Fe^2+^ and ^52^Mn^2+^ data comparison (Figure 6B). Data points from each of the treatment conditions clustered together, indicating behavior within a treatment type that was similar and distinct from other treatment types. Of greatest note, HM053 mutants in the dark showed elevated uptake of ferric and ferrous radiotracer, indicated by the clustering apart from all other treatments and location toward the end of the ferric feature vector and overall positive Y-value. The auxin-deficient mutant, *ipdC*, also showed greatest ferrous radiotracer uptake but this was not observed until exposed to light; otherwise, the *ipdC* mutant clustered near to the FP10 indicating not much difference, other than a slightly elevated ferric uptake, in iron uptake capacity in the dark relative to FP10 in either light condition. The mutant FP10 in both light and dark were nearly indistinguishable from one another and thus, did not change under light stimulus. This demonstrated nicely that the ^59^Fe uptake capacity of the bacteria correlated with BNF capacity of the mutant strain while also showing a light dependency when BNF capacity was available to the bacteria. Additionally, FP10 showed the most similar behavior in comparing ferrous iron and manganese uptake, while *ipdC* showed the most dissimilarity here owing to its significant light dependency noted for ^52^Mn^2+^ uptake.

## 4. Conclusions

As the increased implementation of bacterial inoculants for the enhancement of agricultural crop yield and plant nutrition continues, it becomes more important to fully understand the biological functions and mechanisms of action behind their usefulness. Here, we investigated the light dependencies for two important microbial functions associated with plant growth promotion including BNF and auxin biosynthesis. Although not a phototrophic organism, we discovered that both these functions were light sensitive in *A. brasilense*, being upregulated in light as opposed to darkness. Additionally, we discovered that certain supporting functions underpinning BNF, and auxin biosynthesis, were also light sensitive. Here, we found that microbial uptake of Fe^2+^ as a key element in forming the nitrogen iron protein was light-sensitive, as was ATP, the major energy source for driving BNF and auxin biosynthesis. Consistent with past studies [1], we found that all three strains of *A. brasilense* showed the same growth performance with each other, and no apparent light dependency. Hence, cellular ATP levels and microbial growth were not directly related. This observation is consistent with past studies in *E. coli* which showed that bacterial growth rates were independent of their cellular ATP concentration [58].

While the present work was not able to connect observed light dependencies of *A. brasilense* with its host’s ability to conduct light within the roots, we must wonder why these microorganisms have evolved with traits allowing them to respond to light stimuli. In a time when use of microbial inoculants in agriculture to promote plant growth is becoming a commonplace practice, ways to improve a plant’s microbiota might seek to boost plant light transmission belowground or seek to improve the light sensing capabilities of these microorganisms.

## Figures and Tables

**Figure 1 microorganisms-11-01727-f001:**
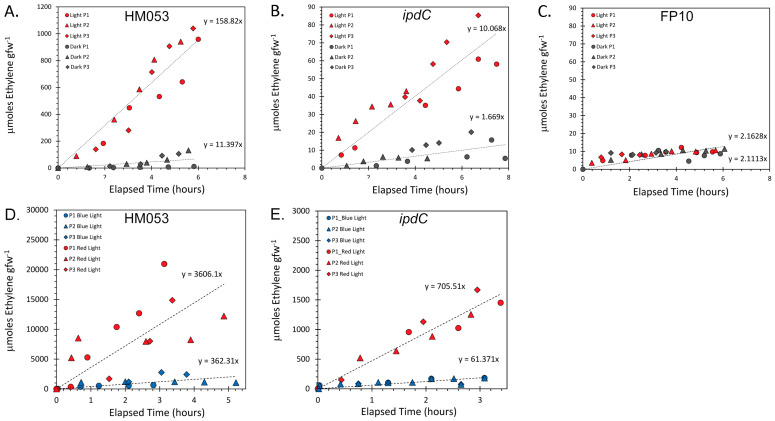
Light dependencies for HM053, *ipdC*, and FP10 strains of *A. brasilense*. Panels (**A**–**C**) represents full spectrum white light (200 µmol m^−2^ s^−1^) versus darkness data collected from ARAs performed on N = 3 plants (labeled P1–P3) that were inoculated with each of the bacteria strains. Panels (**D**,**E**) represent red light versus blue light data collected from ARAs performed on N = 3 plants that were inoculated with either HM053 or *ipdC* bacteria. Light intensities for red and blue light studies were 800 µmol m^−2^ s^−1^, measuring 4 times the intensity of the full spectrum white light.

**Figure 2 microorganisms-11-01727-f002:**
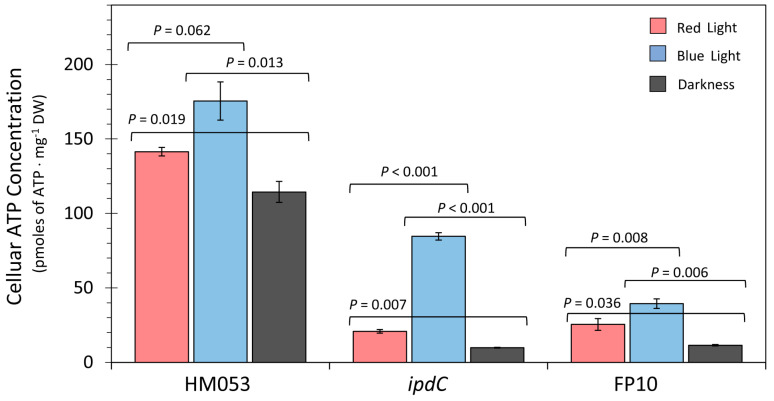
Comparative measurements of cellular ATP concentrations are presented as pmoles of ATP per mg of dry weight (mg^−1^ DW) of extracted bacteria cells for HM053, *ipdC*, and FP10 mutant strains of *A. brasilense* bacteria. Bacteria were grown in liquid cultures at 30 °C for 48 h under red light, blue light, and in darkness. Light intensities were 200 µmol m^−1^ s^−1^. Data bars reflect average values ± SE for N = 6 replicates. Levels of significance are shown by the *p*-values where *p* < 0.05 was statistically significant.

**Figure 3 microorganisms-11-01727-f003:**
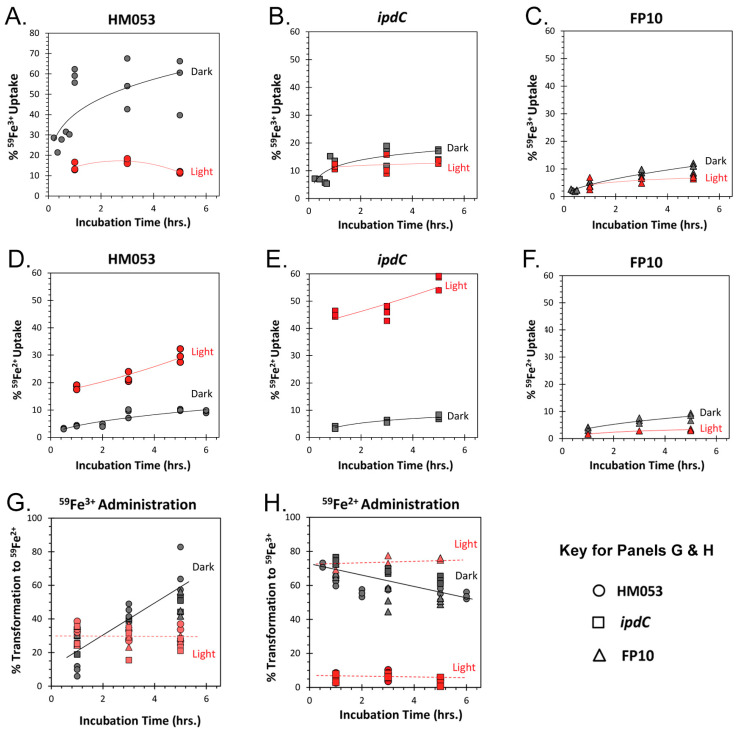
Light dependencies for ^59^Fe uptake as ^59^Fe^3+^ in HM053, *ipdC*, and FP10 bacteria (Panel (**A**–**C**)), and as ^59^Fe^2+^ HM053, *ipdC*, and FP10 bacteria (Panel (**D**–**F**)). Studies were conducted in triplicate for each time point measured extending out to 5 h incubation with radiotracer. Panels (**G**,**H**) also depict how the original oxidation state of the ^59^Fe tracer was metabolically transformed to its other oxidation state over time after being taken up by the bacteria. Data in Panel G represent the percent metabolic transformation of ^59^Fe^3+^-to-^59^Fe^2+^.after administration of ^59^Fe^3+^. Data in Panel H represent the percent metabolic transformation of ^59^Fe^2+^-to-^59^Fe^3+^ after administration of ^59^Fe^2+^.

**Figure 4 microorganisms-11-01727-f004:**
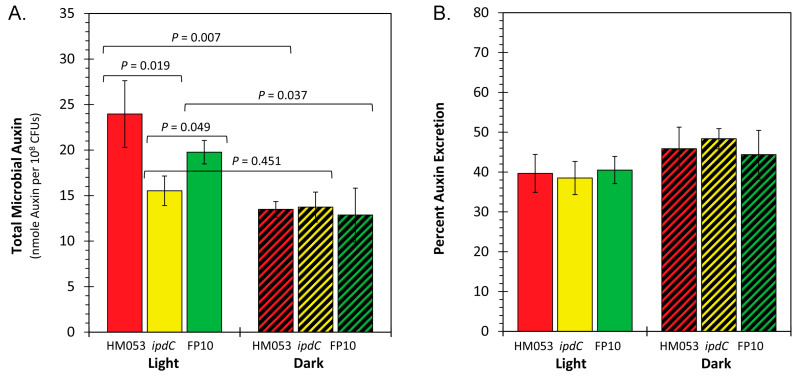
Panel (**A**): total microbial auxin content presented as nmole auxin per 10^8^ colony forming units (CFUs). Treatments included light (equal intensities of red and blue light) versus darkness and examined across the functional mutants of *A. brasilense* including HM053, *ipdC*, and FP10. Data bars reflect average values ± SE for N = 6 replicates. Panel (**B**): relative percent of auxin excreted by the bacteria cells after 48 h of growth under light treatment.

**Figure 5 microorganisms-11-01727-f005:**
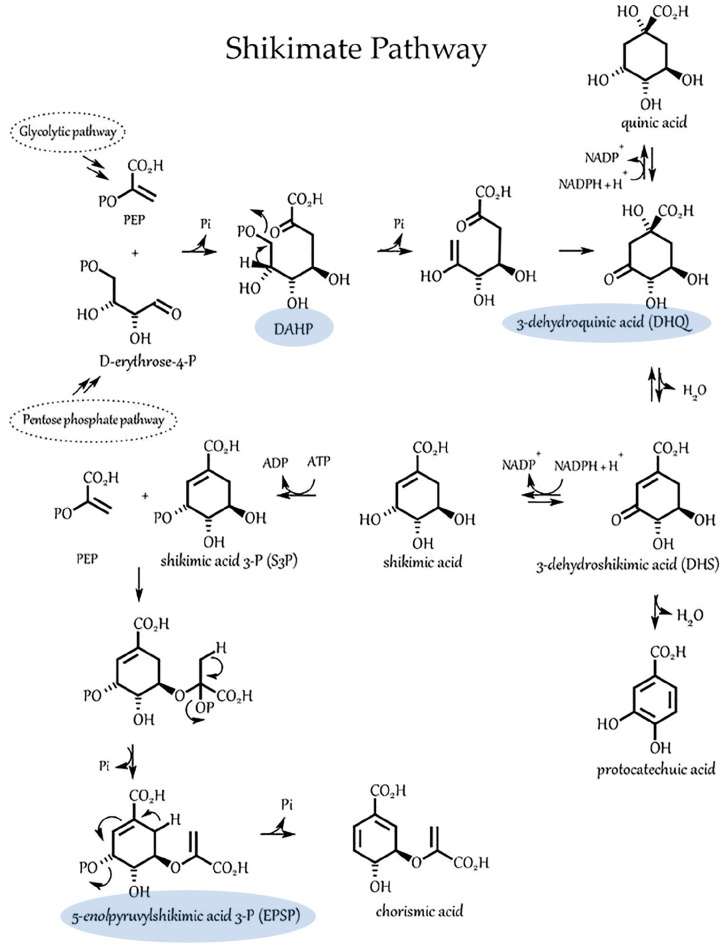
The shikimate pathway common to plants and microorganisms involves seven enzymatic steps that starts with the Aldol condensation of phosphoenol pyruvate (PEP) and erythrose-4-phosphate. At least three of the seven enzymatic steps in this pathway (highlighted in blue) are known to involve divalent manganese.

**Figure 6 microorganisms-11-01727-f006:**
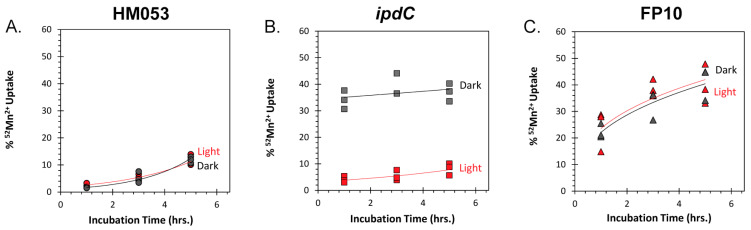
Light dependencies for ^52^Mn^2+^ uptake in HM053, *ipdC*, and FP10 bacteria (Panel (**A**–**C**)). Studies were conducted in triplicate for each time point measured extending out to 5 h incubation with radiotracer.

**Figure 7 microorganisms-11-01727-f007:**
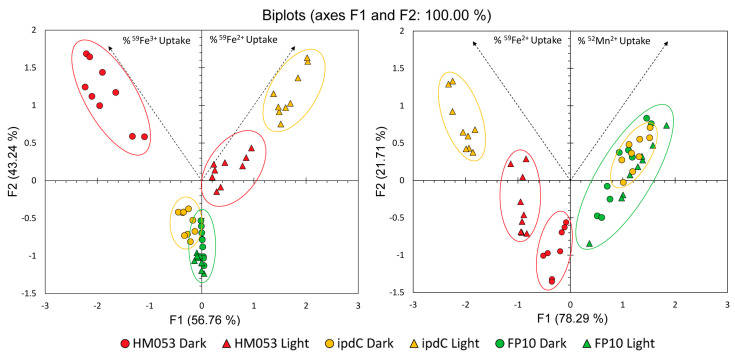
Principle Component Analyses comparing uptake of ^59^Fe^3+^ and ^59^Fe^2+^ (**left panel**), and ^59^Fe^2+^ and ^52^Mn^2+^ (**right panel**) as a function of treatment type (light vs. darkness) and microbial type.

## Data Availability

All data needed to evaluate the conclusions in the paper are present in the main text or the Appendix A.

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
