# Peer review of "Studies Using Mutant Strains of Azospirillum brasilense Reveal That Atmospheric Nitrogen Fixation and Auxin Production Are Light Dependent Processes"

_microorganisms, 2023, doi:10.3390/microorganisms11071727_

Round 1

Reviewer 1 Report

The paper by Bauer Housh et al. reports a very interesting and important study on light-induced processes driven by several Azospirillum brasilence mutant strains (derivatives of the ubiquitous widely studied wild-type strain Sp7). The paper is clearly written, contains a lot of valuable experimental results, is of broad microbiological interest and can provide a good contribution to the journal after some revisions related to the comments listed below.

 (General comment)

The title of the article does not match its content. In the experimental work, mutant strains only were used. The wild-type strain A. brasilense Sp7 was not used in the experiments (at least as a control), and it is unclear how the studied parameters would change in it as compared to the mutants.

What has in fact been studied in the experiments on incubation of biomass with radioactive iron? In the experimental part, it is written that bacteria were first grown on a medium with natural iron, then the culture was collected and incubated with radioactive iron-59 for 1, 3, and 5 hours, i.e. bacterial growth was mostly absent. In this case, there was uptake of those traces of Fe-59 and, at least in part, replacement of natural iron with the Fe-59 taken up, but not the real processes of assimilation. In addition, as is known, iron is contained as part of a large number of bacterial proteins (and enzymes), primarily ferritins, iron-storage proteins. Where the Fe-59, which is taken up, really goes cannot be clarified by these experiments. Thus, the influence of light on the absorption, replacement, and, probably, the change in the oxidation state of radioactive iron, but not on the processes of bacterial Fe assimilation, was in fact studied. In this case, the discussion of iron assimilation is incorrect and should be rewritten accordingly (see also below my comments to lines 313-361 about Fe assimilation in azospirilla).

 Some other comments

To cite those parts of text which are commented in this report, line numbers (indicated in the right-hand margins of the PDF file) in the manuscript, or their range, are used (in parentheses).

 (56-64) In the introduction, for bacteria of the genus Azospirillum, of a dozen of references cited, only a couple are dated within the last years; others are mostly old. For more informativity, a few most recent publications (review papers) on various agricultural aspects of azospirilla, need to be cited and briefly mentioned, e.g.: Cruz-Hernández M.A. et al. Microorganisms (2022) https://doi.org/10.3390/microorganisms10051057 ; Aloo B.N. et al. Front. Plant Sci. (2022) https://doi.org/10.3389/fpls.2022.1002448 ; Cassán F. et al. Biol. Fert. Soils (2020) https://doi.org/10.1007/s00374-020-01463-y ; Fukami J. et al. AMB Expr. (2018) https://doi.org/10.1186/s13568-018-0608-1 (maybe some more recent review papers have appeared, pls check).

 (146) In sec. 2.4 of the Experimental, for the 59Fe preparations used, it is important to specify briefly but clearly the following details:
(a) for the reader to easily assess the amount of 59Fe added (in moles), give an equivalent of its molar content in a 1 mCi dose and also its concentration (in moles per litre) in the experimental tubes (using the value of specific radioactivity of 59Fe; please calculate using the coefficients: 1 mCi = 37 MBq; 1 Bq = 1 radioactive decay of a nucleus per 1 second);
(b) in which chemical form the 59Fe(III) and 59Fe(II) preparations were obtained and added to the experimental solutions with bacteria (if as a complex, with which complexing agent? If as a free ion, how could the possibility for 59Fe(III) hydrolysis and formation of polynuclear hydroxo complexes influence the results and be compared with non-hydrolysing and non-polymerising 59Fe(II) ions?);
(c) how was the possibility of oxidation of 59Fe(II) (ferrous) ions in air prevented or controlled?
(If necessary, add relevant comments also in the discussion of the results.)

 (313-361) While the authors’ experimental data with 59Fe acquisition and transformations in cells of these azospirillum mutants are undoubtedly of great importance, yet the authors have not mentioned the publications on Fe assimilation and, more importantly, its concomitant redox transformations in A. brasilense cells reported for the first time for azospirilla in the following papers: Kamnev et al., Hyperfine Interact. (2014) https://doi.org/10.1007/s10751-013-0929-z ; Alenkina et al., J. Mol. Struct. (2014) https://doi.org/10.1016/j.molstruc.2014.04.084 ; Kamnev et al., Bull. Russ. Acad. Sci. Phys. (2015) https://doi.org/10.3103/S1062873815080110 ; Kovács K. et al., Anal. Bioanal. Chem. (2016) https://doi.org/10.1007/s00216-015-9264-3 ; Kamnev et al., Spectrochim. Acta Part A: Mol. Biomol. Spectrosc. (2020) https://doi.org/10.1016/j.saa.2019.117970 , and summarized in the recent comprehensive review in Russ. Chem. Rev. (2021) https://doi.org/10.1070/RCR5006 (using Mossbauer spectroscopy, particularly in its 57Fe transmission variant). Importantly, it was shown that, while assimilating 57Fe(III)-NTA complex from the medium, different azospirillum strains were able to actively reduce ferric iron to ferrous iron (around ~22% to 33% of the total cellular iron assimilated from the medium after 18 h of growth, i.e. by the end of the logarithmic growth phase, and even to 48% afterwards in one strain, Sp245). These very relevant reports have to be briefly discussed, with the accompanying citations, in relation to the 59Fe(III) and 59Fe(II) assimilation studies for the A. brasilense mutants reported in this manuscript.

 (REFERENCES List)

Please check the whole list for misprints and correct where necessary. Use italics for all Latin names in article titles (e.g. “Arabidopsis thaliana” in [2]; Rhodospirillum rubrum in [19]).

There are some irrelevant references. The authors must use more adequately cited articles. In view of that, it is strongly advised to thoroughly check up and verify the relevance of the cited references (and replace them when necessary). For instance, reference [1] on line 43: this reference is on A. brasilense Sp7 but not on aerobic organisms (as mentioned in the text).

[20] First names of the authors are used instead of surnames (i.e., not Romina M., but Molina R.; not Gaston L., but Lopez G., etc.); see the paper (https://doi.org/10.1007/s00203-020-01829-8) and check their own references in their References list (e.g., in Molina R. et al. (2014), etc.).

[27] In the article title, not “van URK” but “van Urk” (“Urk” is part of a surname).

English is correct.

Author Response

Responses to Reviewer 1 (Author responses are in italics)

The title of the article does not match its content. In the experimental work, mutant strains only were used. The wild-type strain A. brasilense Sp7 was not used in the experiments (at least as a control), and it is unclear how the studied parameters would change in it as compared to the mutants. The title has been changed to “Studies using mutant strains of Azospirillum brasilense reveal that atmospheric nitrogen fixation and auxin production are light dependent processes,” to reflect that we only used mutant strains of A. brasilense in our studies to gain insight on possible light dependencies of the biological processes noted. That said, I would argue that the our inclusion of data on the Sp7 wild-type strain would not have served any purpose as the mutant strains provided us with a complete range of their biological functions.

What has in fact been studied in the experiments on incubation of biomass with radioactive iron? I would like to thank the reviewer for bringing to my attention the extensive background literature on microbial iron assimilation that our paper was lacking. Please see the extensive additional text (Lines 360-371) that includes more background references (see Ref 43-46) on microbial iron uptake.  That description also includes an extensive compilation of cited literature using Mossbauer spectroscopy (reviewed in Ref 47) with specific studies (Ref 47-52) that provided insight into microbial uptake, metabolic transformation, and chemical form within the bacteria cells.  This background places results from our radiotracer uptake studies in better context.

In the experimental part, it is written that bacteria were first grown on a medium with natural iron, then the culture was collected and incubated with radioactive iron-59 for 1, 3, and 5 hours, i.e. bacterial growth was mostly absent. In this case, there was uptake of those traces of Fe-59 and, at least in part, replacement of natural iron with the Fe-59 taken up, but not the real processes of assimilation. In addition, as is known, iron is contained as part of a large number of bacterial proteins (and enzymes), primarily ferritins, iron-storage proteins. Where the Fe-59, which is taken up, really goes cannot be clarified by these experiments. Thus, the influence of light on the absorption, replacement, and, probably, the change in the oxidation state of radioactive iron, but not on the processes of bacterial Fe assimilation, was in fact studied. In this case, the discussion of iron assimilation is incorrect and should be rewritten accordingly (see also below my comments to lines 313-361 about Fe assimilation in azospirilla). I agree with the reviewer here and have replaced the term ‘assimilation’ with ‘uptake’ throughout the paper to make this distinction. Also, I added more detail in our methods section on the nature of our radiotracer studies (i.e. details on molar iron mass and pH of our studies) which is relevant to later comments made by this reviewer regarding ferrous oxidation and ferric ion hydrolysis to insoluble iron hydroxide. Please see lines 186-199 with the inclusion of new Ref 29 addressing conditions for ferric hydrolysis.

Some other comments
(56-64) In the introduction, for bacteria of the genus Azospirillum, of a dozen of references cited, only a couple are dated within the last years; others are mostly old. For more informativity, a few most recent publications (review papers) on various agricultural aspects of azospirilla, need to be cited and briefly mentioned, e.g.: Cruz-Hernández M.A. et al. Microorganisms (2022); Aloo B.N. et al. Front. Plant Sci. (2022); Cassán F. et al. Biol. Fert. Soils (2020); Fukami J. et al. AMB Expr. (2018). Thank you for bringing these more recent papers to my attention.  I have included these more recent works in the Introduction (see Ref 10-13).

 (146) In sec. 2.4 of the Experimental, for the 59Fe preparations used, it is important to specify briefly but clearly the following details:

(a) for the reader to easily assess the amount of 59Fe added (in moles), give an equivalent of its molar content in a 1 mCi dose and also its concentration (in moles per litre) in the experimental tubes (using the value of specific radioactivity of 59Fe; please calculate using the coefficients: 1 mCi = 37 MBq; 1 Bq = 1 radioactive decay of a nucleus per 1 second);  Please see lines 163-164 where the carrier mass of iron is specified. This information is relevant to the ensuing discussion of ferric hydrolysis on lines 186-199.

(b) in which chemical form the 59Fe(III) and 59Fe(II) preparations were obtained and added to the experimental solutions with bacteria (if as a complex, with which complexing agent? If as a free ion, how could the possibility for 59Fe(III) hydrolysis and formation of polynuclear hydroxo complexes influence the results and be compared with non-hydrolyzing and non-polymerising 59Fe(II) ions?); Please see additional text on lines 186-199.  That discussion along with cited paper (Ref 29) supports our statement that ferric hydrolysis under our experimental conditions likely was not a major confounding feature to the radiotracer uptake studies.

(c) how was the possibility of oxidation of 59Fe(II) (ferrous) ions in air prevented or controlled? (If necessary, add relevant comments also in the discussion of the results.) As stated in lines 186-199, the slightly acidic nature of our studies likely minimized ferrous oxidation.

 (313-361) While the authors’ experimental data with 59Fe acquisition and transformations in cells of these Azospirillum mutants are undoubtedly of great importance, yet the authors have not mentioned the publications on Fe assimilation and, more importantly, its concomitant redox transformations in A. brasilense cells reported for the first time for Azospirilla in the following papers: Kamnev et al., Hyperfine Interact. (2014), Alenkina et al., J. Mol. Struct. (2014), Kamnev et al., Bull. Russ. Acad. Sci. Phys. (2015), Kovács K. et al., Anal. Bioanal. Chem. (2016), Kamnev et al., Spectrochim. Acta Part A: Mol. Biomol. Spectrosc. (2020) and summarized in the recent comprehensive review in Russ. Chem. Rev. (2021). Importantly, it was shown that, while assimilating 57Fe(III)-NTA complex from the medium, different azospirillum strains were able to actively reduce ferric iron to ferrous iron (around ~22% to 33% of the total cellular iron assimilated from the medium after 18 h of growth, i.e. by the end of the logarithmic growth phase, and even to 48% afterwards in one strain, Sp245). I would like to thank the Reviewer for pointing out these important works.  All the papers indicated by the Reviewer above have been included in the revised manuscript and our results regarding metabolic transformation related back to these earlier works with our additional discussion found in lines 360-381.  I would also point out that I decided to revise Figure 3G for clarity.  In that revision, I replotted the data as % transformation.  When we administered 59Fe3+ in Fig. 3G, our data now shows on average 30% transformation to the ferric state under illumination regardless of the mutant strain used or incubation time (up to 5 hours). These results are very similar to the Mossbauer data.  However, a very different story unfolds when bacteria are in darkness.  Unfortunately, all the Mossbauer work was conducted in light. Hence, I could relate our new data in darkness to prior work.  Also, our data on ferrous transformation to the ferric state presented in Fig. 3H could not be discussed relative to the prior published Mossbauer data since those earlier works did not examine how ferrous would be assimilated and transformed by these bacteria. However, the features presented in that figure panel are discussed in light of what we have learned about BNF in darkness and microbial ferrous demands to support that function.

These very relevant reports have to be briefly discussed, with the accompanying citations, in relation to the 59Fe(III) and 59Fe(II) assimilation studies for the A. brasilense mutants reported in this manuscript. In the revised submission, our data regarding metabolic transformation of 59Fe brings our work into context with the earlier Mossbauer studies.  

Please check the whole list for misprints and correct where necessary. Use italics for all Latin names in article titles (e.g. “Arabidopsis thaliana” in [2]; Rhodospirillum rubrum in [19]). The references have been checked for proper formatting and corrections made where noted above.

There are some irrelevant references. The authors must use more adequately cited articles. As recommended by the Reviewer more recent publications on PGPB were cited in our revised manuscript (see Ref. 10-13). In view of that, it is strongly advised to thoroughly check and verify the relevance of the cited references (and replace them when necessary). For instance, reference [1] on line 43: this reference is on A. brasilense Sp7 but not on aerobic organisms (as mentioned in the text). I deleted the word ‘aerobic’ from that sentence to make it relevant, but chose to retain Ref 1 in the listing as there was data on microbial growth performance and light that was discussed later in our manuscript.

[20] First names of the authors are used instead of surnames (i.e., not Romina M., but Molina R.; not Gaston L., but Lopez G., etc.); see the paper (https://doi.org/10.1007/s00203-020-01829-8) and check their own references in their References list (e.g., in Molina R. et al. (2014), etc.). References have been corrected.

[27] In the article title, not “van URK” but “van Urk” (“Urk” is part of a surname). Reference has been corrected.

Reviewer 2 Report

Dear authors

I have read your article related to the effect of light on several strains of Azospirillum. I have reviewed it thoroughly, and even when the radioactivity methods are not my field of expertise, I found quite interesting the whole study. I have no comments to add. 

Author Response

Responses to Reviewer 2 (Author responses are in italics)

I have read your article related to the effect of light on several strains of Azospirillum. I have reviewed it thoroughly, and even when the radioactivity methods are not my field of expertise, I found quite interesting the whole study. I have no comments to add. Thank you for your feedback.

Reviewer 3 Report

The article “Biological functions of Azospirillum brasilense that can benefit host growth are responsive to light stimuli” authored by Housh et al. is devoted to study of the reaction of non-phototrophic plant growth promoting bacteria to light stimuli. The response to these stimuli may be due to the presence of phytochromes in A.brasilense, which are able to perceive light in the absence of pigments. The authors studied changes in nitrogenase activity, auxin biosynthesis, ATP biosynthesis, iron and manganese consumption depending on the lighting regime. In general the results seems to be good and valid. But there are several minor revision that should be corrected to improve it.

Minor remarks:

L. 49. Gene name should be given in italic

Should «Mason jar» be capitalized or lowercase? There are different variants in the text.

L.146. Hyphen at the end of the title should be deleted.

L. 202 annihilation, not annhilation

L.326 “and” should be not italic

Figure 5. I am not sure that just a scheme of shikimate pathway should be given in the results section of an experimental article. Perhaps this scheme should be modified. For example, highlight reactions on it with a color that depend on the content of manganese. Or highlight in another way the connection between the shikimate path and the results obtained in this article.

Author Response

Responses to Reviewer 3 (Author responses are in italics)

L.49. Gene name should be given in italic. Correction made.

Should «Mason jar» be capitalized or lowercase? There are different variants in the text. Yes, Mason jar should be capitalized.  The entire manuscript was check to ensure uppercase ‘M’ was used.

L.146. Hyphen at the end of the title should be deleted. Correction made.

L.202. annihilation, not annhilation. Spelling correction made.

L.326. “and” should be not italic.  Italics were removed.

Figure 5. I am not sure that just a scheme of shikimate pathway should be given in the results section of an experimental article. Perhaps this scheme should be modified. For example, highlight reactions on it with a color that depends on the content of manganese. Or highlight in another way the connection between the shikimate path and the results obtained in this article. I feel showing the Figure 5 pathway in the body of the manuscript does add value for those readers not acquainted with the complexity of the shikimate pathway and its numerous steps.  However, I did redo Figure 5 to highlight those steps/enzymes in blue that have been documented to have a dependency on manganese.